# Thermal Fatigue Effect on the Grain Groove Profile in the Case of Diffusion in Thin Polycrystalline Films of Power Electronic Devices

**DOI:** 10.3390/mi14091781

**Published:** 2023-09-17

**Authors:** Tayssir Hamieh, Ali Ibrahim, Zoubir Khatir

**Affiliations:** 1Faculty of Science and Engineering, Maastricht University, P.O. Box 616, 6200 MD Maastricht, The Netherlands; 2Systèmes et Applications des Technologies de l’Information et de l’Energie (SATIE), Gustave Eiffel University, 25 allée des Marronniers, 78000 Versailles, France; ali.ibrahim@univ-eiffel.fr (A.I.); zoubir.khatir@univ-eiffel.fr (Z.K.)

**Keywords:** fourth-order differential equation, diffusion, evaporation, groove, surface energy, thermal fatigue, electronic devices

## Abstract

In a previous paper, we solved the partial differential equation of Mullins’ problem in the case of the evaporation–condensation in electronic devices and gave an exact solution relative to the geometric profile of the grain boundary grooving when materials are submitted to thermal and mechanical solicitation and fatigue effect. In this new research, new modelling of the grain groove profile was proposed and new analytical expressions of the groove profile, the derivative and the groove depth were obtained in the case of diffusion in thin polycrystalline films by the resolution of the fourth differential equation formulated by Mullins that supposed $y^{\prime 2}\ll 1$. The obtained analytical solution gave more accurate information on the geometric characteristics of the groove that were necessary to study the depth and the width of the groove. These new findings will open a new way to study with more accuracy the problem of the evaporation–condensation combined to the diffusion phenomenon on the material surfaces with the help of the analytical solutions.

## 1. Introduction

The thermal fatigue plays an important role during of degradation of interconnection compartments of power electronic devices. The temperature variations resulting from the power cycling has as consequences the stresses and plastic deformations that can affect the microstructure of the materials at the interconnection interfaces of upper metallic parts. Wires and metallization layers more solicited than silicon layers lead to the distortion of material interfaces when the temperature increases, leading to the deformation or degradation of the material surfaces. This will decrease the composite life and leads to an accelerated degradation. The arrangement of grains and grain boundaries is key to understanding the microstructure of metals and composites. When subjected to thermal and mechanical stresses, the variation in surface energies between adjacent grains, confined by the grain boundary, can cause the grains to separate. This phenomenon occurs due to the thermal and mechanical deformation of the grain boundary and the grain groove profile. Such occurrences are commonly observed in the bonding wires utilized in electronic devices.

Some authors [1,2,3] have focused on examining the impact of microstructure and physicochemical properties on degradation processes. In the literature [4,5,6], three effects were investigated. The first two effects examined the influence of bonding procedures and temperature on crack formation and the microstructure of the interconnection zone. Meanwhile, the third effect explored the relationship between material purity, grain size, and hardness during cycling. The metallization layer, typically around 5 μm thick, deposited on the chips undergoes significant distortion compared to materials such as silicon when exposed to high temperature. This distortion results in substantial tensile and compressive stresses, leading to notable inelastic strains [7]. It has been reported that thermomechanical cycling can cause two main types of degradation on the topside of power chips: metallization reconstruction and degradation of bonding contacts [7,8,9]. It is assumed that during cyclic aging, a progressive effect of condensation–evaporation occurs, leading to structural degradation and grooving of the film. However, the precise mechanism of this degradation is not yet fully understood, and further efforts are required to better comprehend the effects of stress parameters on the degradation of contacts between metallization and bond wire. This involves finding a mathematical solution to describe the formation of grain boundary grooving in polycrystalline thin films. Several solutions to this mathematical problem have been proposed in the literature [10,11,12,13,14,15,16,17,18,19,20]. In 1957, Mullins [10] conducted a study on the thermal effect on the profile of grain boundary grooving, laying the foundation for subsequent research on this phenomenon [13,14,15,16,17,18,19,20,21]. Various studies have focused on the development of this phenomenon, particularly exploring evaporation–condensation, surface diffusion, and formulating the mathematical problem that describes the profile of grain boundary grooving [10,11,12]. Some authors [21,22] tried to adapt integrable nonlinear evolution equations related to the well-known linearizable diffusion equation to derive a new integrable nonlinear equation which models the surface evolution of anisotropic material accompanying the action of evaporation–condensation and surface diffusion [22].

A multiple integration technique allowing to solve high-order diffusion equations was proposed by Hristov [23] based on multiple integration procedures by applying the heat-balance integral method of Goodman and the double integration method of Volkov. Hristov [24] presented a solution for the linear diffusion models of Mullins’ thermal grooving [10,11,12]. Fourth-order diffusion equations are commonly encountered in various applications, including surface diffusion on solids [10,11,12,25,26,27,28] and thin film theory [27,28]. Unlike second-order diffusion equations, fourth-order equations generally do not satisfy any known maximum principle. Even with simple time-independent linear boundary conditions, evolving solutions tend to generate additional extrema from initially smooth conditions [29]. Broadbridge [30] studied the problem of a surface groove by evaporation–condensation governed by $\frac{\partial y}{\partial t}=\frac{\frac{\partial^{2}y}{\partial x^{2}}}{1+{\left(\frac{\partial y}{\partial x}\right)}^{2}}$. The depth of a groove at a grain boundary was predicted without any approximation [30]. Chugunova and Taranets [31] studied the initial–boundary value problem associated with the fourth-order Mullins equation with initial data. They considered this problem by assuming that the specific free energy of the boundary is lower than the surface free energy. The Mullins equation, originally introduced by Mullins in 1957 [10], is a model used to analyze the evolution of surface grooves at the grain boundaries of heated polycrystals. Chugunova and Taranets [31] successfully demonstrated the global existence of weak solutions over time and established that the energy minimizing steady state serves as the global attractor. Gurtin and Jabbour [32] developed a regularization theory that incorporates curvature effects, including surface diffusion and bulk–surface interactions. They investigated two specific cases: (i) the interface considered as a boundary between bulk phases or grains, and (ii) the interface between an elastic thin film bonded to a rigid substrate and a vapor phase depositing atoms on the surface [32].

Huang [33] conducted isothermal stress relaxation tests on electroplated Cu thin films, considering both passivated and unpassivated films. Based on a kinetic model, Huang [33] deduced grain-boundary and interface diffusivities and provided numerical and analytical solutions for the coupled diffusion problems. The study also analyzed the impact of surface and interface diffusivities on stress relaxation in polycrystalline thin films, comparing the results to experimental data. Asai and Giga [34] considered the surface diffusion flow equation under specific boundary conditions. The problem of Mullins (1957) was proposed to model the formation of surface grooves on the grain boundaries, where the second boundary condition $y^{\prime \prime \prime }\left(0\right)=0$ is replaced by zero slope condition on the curvature of the graph. Asai and Giga solved the initial–boundary problem with homogeneous initial data for construction of a self-similar solution and a solution was proposed by using a semidivergence structure. Escher et al. [35] demonstrated the existence and uniqueness of classical solutions for the motion of immersed hypersurfaces driven by surface diffusion. They focused the surface diffusion proposed by Mullins [10,11,12] to model surface dynamics for phase interfaces when the evolution is governed solely by mass diffusion within the interface. Other studies were devoted to the diffusion problems, grain boundary migration, and grain dynamics evolution in materials [36,37,38,39,40,41,42].

Mullins et al. [43] have linearized the differential equation by assuming a very small slope at any point of the grain profile. In 1975, Brailsford and Gjostein [44] derived approximate solutions by studying the influence of surface energy anisotropy on morphological changes occurring by surface diffusion on simply shaped bodies. Wherever a grain boundary intersects the surface of a polycrystalline material, a groove develops. At the root of the groove, a balance between grain–boundary tension and surface tension produces an equilibrium angle [45]. The difference in chemical potential between the curved surface near the groove’s root and the smoother surface farther away results in material drift. Tritscher [46] considered the boundary–value problem concerning the formation of a single groove due to surface diffusion at the junction of a bicrystal, assuming that the grain boundary remains planar.

Martin [47] extended the original Mullins theory of surface grooving due to a single interface to multiple interacting grooves formed by closely spaced flat interfaces. Martin considered two cases: the first involved simplifying Mullins’ analysis using Fourier cosine transforms instead of Laplace transforms, while the second dealt with an infinite periodic row of grooves. Martin [40] also solved the problem for two interacting grooves. Analytical solutions for the fourth partial differential equation governing the groove profile in metals have not been found in the literature.

In a previous study [48], we addressed the mathematical problem associated with the second nonlinear partial differential equation in Mullin’s problem. We focused on the case of the evaporation–condensation and provided an exact solution for the geometric profile of grain boundary grooving when materials are subjected to thermal and mechanical stress, as well as fatigue effects.

This paper is devoted to model the grain groove profile governed by the fourth-order partial differential equation in the case of diffusion in thin polycrystalline films. An analytical and exact solution to the Mullins approximated problem, $\frac{\partial y}{\partial t}+B\frac{\partial^{4}y}{\partial x^{4}}=0$, was given.

## 2. Mathematical Formulation in the Diffusion Case

In this section, we were interested to the derivation of the differential equation that describes the evolution of a two-dimensional surface of small slope under capillary driving forces and surface diffusion transport. Surface properties are assumed to be independent of orientation. For a point on the surface at which the mean curvature is *c*, the chemical potential $\mu \left(c\right)$ per atom can be written as
(1)$$\mu \left(c\right)=\mu_{0}+\gamma \omega c$$
where $\mu_{0}$ is the chemical potential of reference for a flat surface (c=0), γ is the surface tension of the metal/vapor interface and ω is the atomic volume of the film material. A gradient of surface curvature will therefore create a gradient of the chemical potential μ, which will produce a drift of atoms on the surface with an average velocity v given by the Nernst–Einstein relation.
(2)$$v=-\frac{D_{s}}{kT}\frac{\partial \mu }{\partial s}$$
or
(3)$$v=-\frac{D_{s}\gamma \omega }{kT}\frac{\partial c}{\partial s}$$
where $D_{s}$ is the surface diffusivity, k is the Boltzmann constant and T the absolute temperature.

The surface current of atoms $J_{S}$ is defined by the product of the average velocity v by the atom number $N_{S}$ per unit surface area s, it is given by the following equation:(4)$$J_{S}=v\;N_{S}$$
(5)$$J_{S}=-\frac{D_{s}}{kT}\frac{\partial \mu }{\partial s}=-\frac{D_{s}\gamma \omega N_{S}}{kT}\frac{\partial c}{\partial s}$$

The evolution of the surface may finally be described by the speed of movement $v_{n}$, of the surface element along its normal:(6)$${v_{n}=-\omega \;\nabla_{s}\;J}_{S}=\frac{D_{s}\gamma \omega^{2}N_{S}}{kT}\frac{\partial^{2}c}{\partial s^{2}}$$
(7)$$v_{n}=B\frac{\partial^{2}c}{\partial s^{2}}$$

Notice that $N_{S}$ is the number of diffusing atoms per unit area, $J_{S}$ the surface current of atoms and B a rate constant given by the following equation:(8)$$B=\frac{D_{s}\gamma \omega^{2}N_{S}}{kT}$$

Equation (7) can be written in the general case as:(9)$$v_{n}=B\;\nabla_{s}^{2}c$$

Equation (9) is the general case for the normal direction velocity, where c is the curvature defined by Equation (10), and *y* is the coordinate of a point at the surface along the axis normal to the initial flat surface. The calculations (see Appendix A) led to the following general diffusion equation (Equation (11)) with the boundary conditions given by Equation (12).
(10)$$c=-\frac{\frac{\partial^{2}y}{\partial x^{2}}}{{\left[1+{\left(\frac{\partial y}{\partial x}\right)}^{2}\right]}^{3/2}}$$
(11)$$\frac{\partial y}{\partial t}=-B\;\frac{\partial }{\partial x}\left[{\left(1+y^{\prime 2}\right)}^{-1/2}\;\frac{\partial }{\partial x}\left[\frac{y^{\prime \prime }}{{\left(1+y^{\prime 2}\right)}^{3/2}}\right]\right]$$
(12)$$\left\{\begin{array}{l} \;\;\;\;y\left(x,\;0\right)=0\\ y\left(0,\;t\right)=-\frac{m\;{\left(Bt\right)}^{\frac{1}{4}}}{\sqrt{2}\Gamma \left(5/4\right)}\\ \;\;\;\;y^{\prime }\left(0,\;t\right)=\tan\theta =m\\ \;\;\;\;\underset{x\to \infty }{\lim}y^{\prime }\left(x,\;t\right)=0\\ \;\;\;\;\underset{x\to \infty }{\lim}y^{\prime \prime }\left(x,\;t\right)=0\\ \;\;\;\;y^{\prime \prime \prime }\left(0,\;t\right)=0 \end{array}\right.$$

## 3. New Study of Mullins’s Case

By adopting a change in variables and defining a new function *g*, as shown by Equation (13), one obtains the equation for the diffusion case. If we suppose a second order approximation of the derivative, $y^{\prime 2}\;\ll 1$, it is easy to deduce the approximated differential equation of Mullins’s case given by Equation (14) (see Appendix A for the full derivation).
(13)$$\left\{\begin{array}{l} y\left(x,\;t\right)=m\;{\left(Bt\right)}^{1/4\;}g\left[\frac{x}{{\left(Bt\right)}^{1/4\;}}\right]\;\\ u\left(x,\;t\right)=\frac{x}{{\left(Bt\right)}^{1/4\;}}\\ y\left(u,\;t\right)=m\;{\left(Bt\right)}^{1/4\;}g\left(u\right) \end{array}\right.$$
(14)$$g^{⁗}-\frac{1}{4}ug^{\prime }+\frac{1}{4}g=0\;$$

With the new boundary conditions:(15)$$\left\{\begin{array}{l} g\left(u,\;0\right)=0\\ g\left(0,\;t\right)=-\frac{1}{\sqrt{2}\Gamma \left(5/4\right)}\\ \underset{u\to \infty }{\lim}g^{\prime }\left(u,\;t\right)=0\\ \underset{u\to \infty }{\lim}g^{\prime \prime }\left(u,\;t\right)=0\\ g^{\prime \prime \prime }\left(0,\;t\right)=0 \end{array}\right.$$

### 3.1. Exact Resolution of Mullins’ Problem

In order to give the exact solution of Mullin’s problem we propose a new method in which a function r is introduced given by Equations (16) and (17).
(16)$$r^{4}-\frac{1}{4}u\;r+\frac{1}{4}=0\;$$
(17)$${r^{4}-\frac{1}{4}u\;r+\frac{1}{4}=\left(r^{2}+\lambda \right)}^{2}-\left(\frac{{8\lambda r}^{2}+u\;r+{4\lambda }^{2}-1}{4}\right)$$

The treatment of these equations will lead to the discriminant delta ${∆}_{\lambda }$ and a particular value for $u=u_{0}=\frac{2^{5/2}}{3^{3/4}}$. Two cases arise for (1) ${∆}_{\lambda }\geq 0$, $u\geq u_{0}$ and (2) ${∆}_{\lambda }\leq 0$, $u\leq u_{0}$. After applying the proper boundary conditions for each case and solving for the unknown problem parameters, these two cases will give us two final analytical expressions for the function $g\left(u\right)$ and the final closed form expression for the profile variation in the grove (see Appendix B for the detailed derivation).

The analytical solution of the fourth order differential equation (Equations (14) and (15)) is finally given by Equation (18).
(18)$$g\left(u\right)=\left\{\begin{array}{c} g_{1}\left(u\right)\;\;\;\;\;\;\;\;\;\;\;for\;u\leq \frac{2^{5/2}}{3^{3/4}}\;\\ g_{2}\left(u\right)\;\;\;\;\;\;\;\;\;\;for\;u\geq \frac{2^{5/2}}{3^{3/4}} \end{array}\right.$$

With:(19)$$\left\{\begin{array}{l} g_{1}\left(u\right)=e^{-p_{1}(u)}\;\left(A_{11}cos\;q_{1}(u)+A_{21}sin\;q_{1}(u)\right)\;\\ g_{2}\left(u\right)=e^{{-p}_{2}(u)}\left(A_{12}cos\;q_{2}(u)+A_{22}sin\;q_{2}(u)\right)\; \end{array}\right.$$

One proved that all parameters and derivatives for the two functions $g_{1}$ and $g_{2}$ are equal and the continuity of the solution and its derivatives is assured at this point $u_{0}$ and consequently at any point of the interval [0, ∞]. The constants of the problem are given by Equation (20):(20)$$\left\{\begin{array}{l} A_{11}=A_{12}=-\frac{1}{\sqrt{2}\times \Gamma \left(5/4\right)}=g_{0}\\ A_{21}=A_{22}=\frac{1}{\sqrt{2}\times \Gamma \left(5/4\right)}={-g}_{0} \end{array}\right.$$

The expressions of the variables $p_{1}(u)$, $q_{1}(u)$, $p_{2}(u)$ and $q_{2}(u)$ are given in Appendix B.

By using the variables $x={\left(Bt\right)}^{1/4}u\left(x,t\right)$ and $y\left(u,t\right)=m{\left(Bt\right)}^{1/4}g\left(u\right)$, the analytical solution $y\left(x,\;t\right)$ can be written as:(21)$$y\left(x,\;t\right)=\frac{m\;{\left(Bt\right)}^{1/4\;}}{\sqrt{2}\times \Gamma \left(5/4\right)}e^{-p[\frac{x}{{\left(Bt\right)}^{1/4\;}}]}\;\left[-cos\;q[\frac{x}{{\left(Bt\right)}^{1/4\;}}]+sin\;q[\frac{x}{{\left(Bt\right)}^{1/4\;}}]\right]$$

### 3.2. Profile of the Groove Shape in the Diffusion Case

The variations in the profile $y\left(x,\;t\right)$ as a function of the distance *x* from the symmetric axis of the groove are plotted on Figure 1.

The study of the solution $y\left(x,\;t\right)$ reveals a damped sinusoidal profile of the groove with an infinity of maxima, minima, and zeros of the solutions. The oscillations can be easily observed in our solution. Mullins mentioned that it is questionable, however, that these oscillations could be observed due to the progressively decreasing amplitude of g. Here, we proved the superiority of our analytical solution that can predict the oscillations, their amplitudes, the zero, the maxima and minima of the groove profile.

As example, we gave on Table 1 the 12 first values of the groove shape parameters and on Table 2 the distance between two consecutive maxima and minima for the first 12 numbers.

We observed that $y_{Max}$ decreases towards zero when *x* increases to the infinity as well as the absolute value of $y_{min}$ (Table 1). This will decrease the distance between two consecutive maxima and minima when the distance x increases. From the first number of optima, on observed on Table 2 that a constant value of $\left|{∆lny}_{Max\;or\;min}\right|$ equal to 3.63 was found for all minima and maxima, whereas the difference $∆x_{Max\;or\;min}$ decreases for the minima and maxima to tend to zero at the infinity.

Our calculations led to draw the curves of Figure 2:

The results of Table 2 and the curves of Figure 2 allowed to give the interpolating equations (Table 3):

Equations given in Table 3 showed the properties of damped sinusoidal functions and the pseudo-periodicity of the various groove parameters and the strong correlations between them showing at the same time the infinity of the number of these different parameters.

On Table 4, we gave the various results obtained by our analytical solution and the Mullins’s results.

The parabolic approximation of the groove profile obtained by Mullins was valid for 0≤x≤1, whereas our approximation more precise is valid for 0≤x≤2.40 (from the origin until the first maximum of the groove shape). On the other hand, the error committed by Mullins’ calculations on the abscissa of the first maximum the zero of the function y and the first inflexion point is about 7%, while that on the ordinate of the profile maximum exceeds 25%. On Table 4, we were able, on the contrary of Mullins’ results, to give more information on the various maxima, minima, zeros, and positive and negative inflexion points of the grove shape profile.

Equation (22) gives the expressions of the parameters $h_{Max}$ and $h_{min}$ representing the depths of the groove taken from the bottom of the grove, respectively, to its first maximum and minimum.
(22)$$\left\{\begin{array}{l} h_{Max}=\varepsilon_{0}+y_{Max.1}\\ h_{min}=\varepsilon_{0}+y_{min.1} \end{array}\right.$$

Now, knowing that
$$\varepsilon_{0}=\frac{m{\left(Bt\right)}^{1/4\;}}{\sqrt{2}\times \Gamma \left(5/4\right)}$$
and
$$\left\{\begin{array}{l} y_{Max.1}=0.260\times m{\left(Bt\right)}^{1/4}\\ y_{min.1}=-0.040\times m{\left(Bt\right)}^{1/4} \end{array}\right.$$

One deduced:(23)$$\left\{\begin{array}{l} h_{Max}=\left[\frac{1}{\sqrt{2}\times \Gamma \left(5/4\right)}+0.260\right]m{\left(Bt\right)}^{1/4}\\ h_{min}=\left[\frac{1}{\sqrt{2}\times \Gamma \left(5/4\right)}-0.040\right]m{\left(Bt\right)}^{1/4} \end{array}\right.$$
and
(24)$$\left\{\begin{array}{c} h_{Max}=1.040\times m{\left(Bt\right)}^{1/4\;}\;\;\;\\ h_{min}=0.740\times m{\left(Bt\right)}^{1/4\;}\;\;\;\; \end{array}\right.$$

The separation distance between two consecutive maxima $d_{Max}$ or minima $d_{min}$ given in Table 5 proved an important variation in this distance as a function of optima number N. Table 5 gave their interpolated expressions.

Table 5 clearly showed that the ratio d/h is independent from the time but it depends on the slope *m*, for example, we can give this ratio for the first maximum (Equation (25)):(25)$$\frac{d_{Max}}{h_{Max}}=\frac{5.02}{m}$$

On Table 6, we presented a comparison between some important parameters obtained by our analytical solution and that of Mullins.

Table 6 showed a certain deviation of Mullins’ results with respect to those of the analytical solution proposed in this paper, that can reach 12% in the case of the first maximum of the groove shape. However, Mullins did not give any additional information on the other maxima, minima, and zeros of the solution and the various inflexion points, while our solution gave more complete information on the different parameters of the groove and also proposed many correlations that can be very useful for the readers.

Here, some information on the coordinates of the positive and negative inflexion points are given on Table 7.

### 3.3. Competition between Evaporation and Diffusion

When studying the evolution of grain boundary groove profiles in the cases of the evaporation/condensation and surface diffusion, Mullins [10] assumed that: (1) the surface diffusivity and the surface energy, $\gamma_{SV}$, were independent of the crystallographic orientation of the adjacent grains and (2) the tangent of the groove root angle, γ, is small compared to unity. Mullins also supposed an isotropic material. The assumption (tan*θ* << 1) was used in all papers’ Mullins to simplify the study of the mathematical partial differential equation. The polycrystalline metal was supposed (3) in quasi-equilibrium with its vapor. The interface properties do not depend on the orientation relative to the adjacent crystals. The grooving process was described by Mullins using the macroscopic concepts (4) of surface curvature and surface free energy. The matter flow (5) is neglected out of the grain surface boundary.

The mathematical equation governing the evaporation–condensation problem can be written here as:(26)$$\frac{\partial y}{\partial t}=C(T)\frac{y^{\prime \prime }\;\left(x\right)}{\left(1+{y^{\prime }\left(x\right)}^{2}\right)}$$
where *C*(*T*) a constant of the problem depending on the temperature *T*, given by Equation (27).
(27)$$C\left(T\right)=\mu \frac{P_{0}\left(T\right)\;\gamma \left(T\right)\;\omega^{2}}{\sqrt{2\pi mkT}}$$
where *γ* is the isotropic surface energy, $P_{0}\left(T\right)\;$ the vapor pressure at temperature *T* in equilibrium with the plane surface of the metal characterized by a curvature *c* = 0, *ω* is the atomic volume, *m* is molecular mass, *μ* the coefficient of evaporation and *k* is the Boltzmann constant.

We remember here the analytical solution of the evaporation case without any approximation [48] given by
(28)$$y\left(x,\;t\right)=\int_{\infty }^{x/2\sqrt{Ct}}\;\frac{\;sin\;\theta }{\sqrt{e^{v^{2}/(2Ct)}-{sin}^{2}\theta }}\;dv$$
and
(29)$$y\left(x,\;t\right)=-\sqrt{\pi Ct}\;sin\;\theta \left[erfc\left(\frac{x}{2\sqrt{Ct}}\right)+\sum_{n=1}^{\infty }\frac{\left(2n\right)!}{{\left(n!\right)}^{2}2^{2n}\;\sqrt{3n}}{\;sin}^{2n}\theta \;\left(erfc\left(\;\frac{x\sqrt{3n}}{2\sqrt{Ct}}\right)\right)\right]$$

By combining the two phenomena of diffusion and evaporation/condensation, one writes:(30)$$\frac{\partial y}{\partial t}=C\frac{y^{\prime \prime }}{\left(1+y^{\prime 2}\right)}-B\;\frac{\partial }{\partial x}\left[{\left(1+y^{\prime 2}\right)}^{-1/2}\;\frac{\partial }{\partial x}\left[\frac{y^{\prime \prime }}{{\left(1+y^{\prime 2}\right)}^{3/2}}\right]\right]$$

With the approximation postulated by Mullins supposing that $y^{\prime 2}\ll 1$ one can write Equation (31).
(31)$$\frac{\partial y}{\partial t}=Cy^{\prime \prime }-By^{⁗}$$

With the constants B and C given by Equation (32).
(32)$$\left\{\begin{array}{l} B=\frac{D_{s}\gamma \omega^{2}N_{S}}{kT}\\ C=\mu \frac{P_{0}\;\gamma \;\omega^{2}}{\sqrt{2\pi m}{\left(kT\right)}^{3/2}} \end{array}\right.$$

Let us put B the profile area. One can write the rate of change in profile area:(33)$$\frac{dB}{dt}=\int_{-\infty }^{+\infty }\frac{\partial y}{\partial t}dx=2\int_{0}^{+\infty }\left[Cy^{\prime \prime }-By^{⁗}\right]dx$$

One writes Equation (34).
(34)$$\frac{dB}{dt}=-2\;\left[Cy^{\prime }\left(0\right)-By^{\prime \prime \prime }\left(0\right)\right]$$

In a previous paper [48], we studied the case of evaporation without this approximation and obtained at the origin Equation (35).
(35)$$\left\{\begin{array}{c} y^{\prime }\left(0,\;t\right)=tan\;\theta =m\;\\ y^{\prime \prime \prime }\left(0,\;t\right)=-2\;m\;\left(1+{\;m}^{2}\right) \end{array}\right.$$

In such case, one obtains Equation (35).
(36)$$\frac{dB}{dt}=-2m\left[C+2B\;\left(1+{\;m}^{2}\right)\right]$$
and therefore, the profile area B of as a function of the slope m and the time t (Equation (37)).
(37)$$B=-2m\left[C+2B\;\left(1+{\;m}^{2}\right)\right]t$$

Equation (37) provides clear evidence that the rate of change in the profile area is influenced by both evaporation and diffusion, contrary to Mullin’s prediction which states that B=−2mC and is independent of surface diffusion.

The profile area A from below to above the original surface is defined by Equation (38).
(38)$$A=-\int_{0}^{x_{0}}y\left(x\right)dx=-m{\left(Bt\right)}^{3/4}\int_{0}^{u_{0}}g\left(u\right)du$$

The calculations of A, detailed in Appendix C, led Equation (39).
(39)$$A=2m{\left(Bt\right)}^{3/4}g^{\prime \prime \prime }\left(u_{0}\right)$$

By considering σ as a new parameter defining the profile area transferred from below to above of the original surface by surface diffusion alone divided by the profile area lost by evaporation, one can write Equation (40).
(40)$$\sigma =\frac{A}{B}=\frac{-2{m{\left(Bt\right)}^{1/2}g}^{\prime \prime \prime }\left(u_{0}\right)}{2m\left[C+2B\;\left(1+{\;m}^{2}\right)\right]t}$$

With $u_{0}=1.22$, our solution giving $g^{\prime \prime \prime }\left(u_{0}\right)=-0.1543$ led us to deduce Equation (41)
(41)$$\sigma =\frac{0.1543\times B^{1/2}t^{-1/2}}{C+2B\;\left(1+{\;m}^{2}\right)}$$

If we suppose that the contact angle is small or ${\;m}^{2}\ll 1$ (for θ<18°) we obtain:(42)$$\sigma =\frac{0.1543\times B^{1/2}}{C+2B\;}t^{-1/2}$$
and therefore the final expression of σ:(43)$$\sigma =\frac{0.1543\times kT{\left({2\pi mD}_{s}N_{S}\right)}^{1/2}}{\omega \gamma^{1/2}\left[\mu P_{0}+2D_{s}N_{S}{\left(2\pi mkT\right)}^{1/2}\right]\;}t^{-1/2}$$

Equation (43) derived from our analytical solution proved that σ depends not only on the time but also on the temperature, at contrary of the relation obtained by Mullins (Equation (44))
(44)$$\sigma =0.38\frac{{\left({2\pi mD}_{s}N_{S}\right)}^{1/2}}{\omega \gamma^{1/2}P_{0}\;}t^{-1/2}$$

Indeed, in the Mullins’s relation (Equation (44)), there is no direct effect of the temperature. To compare between the two previous expressions (43) and (44), we calculated the ratio of the two values σ obtained by our solution $\left(\sigma_{H}\right)$ and that of Mullins $\left(\sigma_{M}\right)$. One obtained Equation (45).
(45)$$\frac{\sigma_{H}}{\sigma_{M}\;}=0.406$$

The ratio $\frac{\sigma_{H}}{\sigma_{M}\;}$ given by Equation (45) clearly indicated an overestimation of the value proposed by Mullins compared to the exact solution.

## 4. Comparison with Experimental Data

In this section, we used the experimental data obtained in the case of two used common metals such as gold and magnesium. On Table 8, we presented the thermodynamic parameters of Au and Mg.

In order to compare between our theoretical results and that obtained by Mullins, we gave on Table 9 the calculated values of *B*, *C*, and σ the two methods for *Au* and *Mg* metals.

We observed that the profile areas corresponding to Au and Mg are overestimated by Mullins’ method (about 2.5 times greater than our new values). On the other hand, the calculated ratio $\frac{\sigma_{Au}}{\sigma_{Mg}}$ of the profile area lost by evaporation of Au and Mg is equal to:(46)$$\frac{\sigma_{Au}}{\sigma_{Mg}}=1.8\times 10^{5}$$

Equation (46) proved that whatever the time, the evaporation of *Au* is $1.8\times 10^{5}$ times more important than that of *Mg*. However, the diffusion of *Mg* particles is greater than that of *Au*.

The same procedure was extended to other usual metals to determine the values of the profile area lost by evaporation. The experimental data for several metals (*Cu*, *Al*, *Sr*, *Li*, *Cs*, *Ti*, *Co*, *Ga*, and *Tl*) were given on Table 10.

These interesting results of Table 10 allowed to classify the various metals by following the increasing order of the profile area:*Cu* < *Al* < *Sr* < *Li* < *Cs* < *Ti* < *Co* < *Ga* < *Tl*

On Table 11, we gave the obtained values of the two constants *C* and *B* of evaporation and diffusion for the different metals.

The constant of evaporation *C* decreases from the cobalt element Co to cesium by respecting the following increasing order:*Co* < *Ti* < *Ga* < *Li* < *Tl* < *Al* < *Cu* < *Sr* < *Cs*

Whereas, this order changes for the constant of diffusion that increases from Cu to Cs with the following order:*Cu* < *Co* < *Ga* < *Li* < *Ti* < *Al* < *Tl* < *Sr* < *Cs*

Another important conclusion concerns the larger value of constant *C* with respect to *B*. It is shown that the value of *C* is about 10^12^ times greater that of *B*. This led to conclude that the diffusion can be in general neglected relative to evaporation.

### 4.1. Consequence of Theoretical Results on the Depth of the Groove

In many experiments, it was proved that the depth groove can vary from 0.1 mm to several 10 mm in the case of diffusion depending on the metal thermal properties and on the width of the groove. In order to understand the thermal behavior of diffusion of the various elements, let us take the typical example where m = 0.20 and calculate the corresponding depth $h_{Max}$ of the groove for metals. The obtained results were given on Table 12.

The results of Table 12 allowed to compare between the depth $h_{Max}$ of the groove for the various metals that can be therefore classified in increasing order of the depth:Cu < Co < Ga < Li = Ti < Al < Tl < Sr < Cs

This result confirmed that previously obtained by the diffusion constant B.

Knowing that the width $w_{Max}$ of the groove is given by Equation (47)
(47)$$w_{Max}=2x_{Max}=4.8\times {\left(Bt\right)}^{1/4\;}$$

One deduced the value of $w_{Max}$ for the different metals presented on Table 13.

### 4.2. Consequences of the New Solution on the Thermodynamic Parameters

The experimental study of the geometric characteristics of the groove for metals can lead to the determination of the two constants of evaporation and diffusion. Indeed, the evaporation constant can be obtained by determining experimentally the value of the profile area B and by considering in first approximation B=−2mCt or $C=-\frac{B}{2mt}$. By determining the value of *C*, it becomes possible to determine the surface energy *γ* of the metal using the relation of the evaporation constant, resulting in the following expression:(48)$$\gamma =\frac{C\sqrt{2\pi m}{\left(kT\right)}^{3/2}}{P_{0}\;\;\omega^{2}}=-\sqrt{\frac{\pi {\left(kT\right)}^{3}}{2m}}\frac{B}{P_{0}\omega^{2}t}$$

The evaluation of the width $w_{Max}$ of the groove will give the value of diffusion constant B by using Equation (47), and therefore:(49)$$B=1.88\times 10^{-3}\frac{w_{Max}^{4}}{t}$$

Knowing *γ* and $w_{Max}$, we will be able to obtain the value of the surface diffusion $D_{s}$:(50)$$D_{s}=2.6\times 10^{-26}\frac{Tw_{Max}^{4}}{\gamma \omega^{2}N_{s}t}$$

### 4.3. Validity of the Approximation of $y^{\prime 2}\ll 1$

Let us consider the case of copper metal to test the validity of $y^{\prime 2}\ll 1$ and draw on Figure 3 the variations in $y^{\prime 2}$ as a function of the distance *x* for different contact angles.

Figure 3 showed that for θ < 30°, the value of $y^{\prime 2}$ < 0.2 and can be approximately neglected behind 1 following Mullins’ approximation. Therefore, for θ > 30°, the approximated fourth partial differential equation proposed by Mullins cannot be used for the diffusion case and then it will be necessary to resolve the non-linear partial fourth order differential equation that cannot be analytically obtained.

### 4.4. Variations in the Groove Profile y(x) and the Derivative y′(x) as a Function of the Distance x of Cu

We used the results of our analytical solution to determine the groove profile and its derivative in the case of copper metal. On Figure 4, we drew the variations in the profile *y*(*x*) *and y′*(*x*) in the case of Cu by noting the geometric parameters of the groove such as $h_{Max}$, $d_{Max}$, and $w_{Max}$. By using our solution, we obtained the following geometric characteristics of the groove:$$h_{Max}=2.16\;\mu m;\;d_{Max}=29.54\;\mu m;\;\frac{w_{Max}}{2}=13.68\;\mu m$$

On Figure 5, we plotted the variations in the profile *y*(*x*) of the groove of *Cu* as a function of the distance *x* for different values of contact angles.

Figure 5 clearly showed the effect of the contact angle of the grove. The groove depth increases when m increases. However, the other characteristics such as $d_{Max}$ and $w_{Max}$ remain the same.

The obtained analytical solution allowed to compare between the groove profiles among various metals. Figure 6 showed different groove characteristics in different metals. It can be seen that the groove depth and the distance between two maxima increased from *Cu* to *Cs* (Figure 6).

Figure 6 also showed the large difference in the behavior of the various metals. The grove phenomenon is more accentuated for *Cs*, whereas *Cu* is less affected by the surface diffusion.

## 5. Conclusions

In this study, we derived an exact solution to the partial differential equation $\frac{\partial y}{\partial t}+By^{⁗}=0$. The obtained solution reveals a damped sinusoidal groove profile in the case of electronic power devices. We provided expressions of zeros, minima, and maxima of the profile as a function of the order number, as well as detailed information about the groove profile y(x) and its derivatives. A comprehensive comparison with Mullins’ results was conducted, demonstrating that Mullins’ predictions significantly overestimate the geometric characteristics of the groove, exceeding the actual values by more than 2.5 times. Additionally, valuable insights into the diffusion behavior of various metals gained through this study. The expressions for the evaporation and diffusion constants and coefficients were also derived, accounting for the groove parameters. New expressions of the profile area lost by evaporation, the surface energy and the surface diffusion coefficients were also obtained. The combination between our new analytical solution and the experimental data of several metals such as *Cu*, *Al*, *Sr*, *Li*, *Cs*, *Ti*, *Co*, *Ga*, and *Tl* gave the geometric parameters such as the depth $h_{Max}$ and the width $w_{Max}$ of the groove in the case of diffusion in these metals, and allowed an interesting comparison between the diffusion in metals as a function of time.

## Figures and Tables

**Figure 1 micromachines-14-01781-f001:**
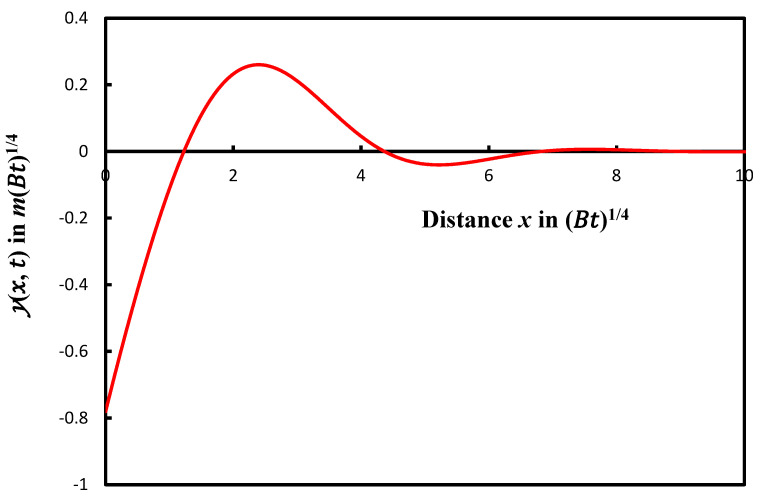
Groove profile giving $y\left(x,\;t\right)$ as a function of the distance from the symmetric axis of the groove.

**Figure 2 micromachines-14-01781-f002:**
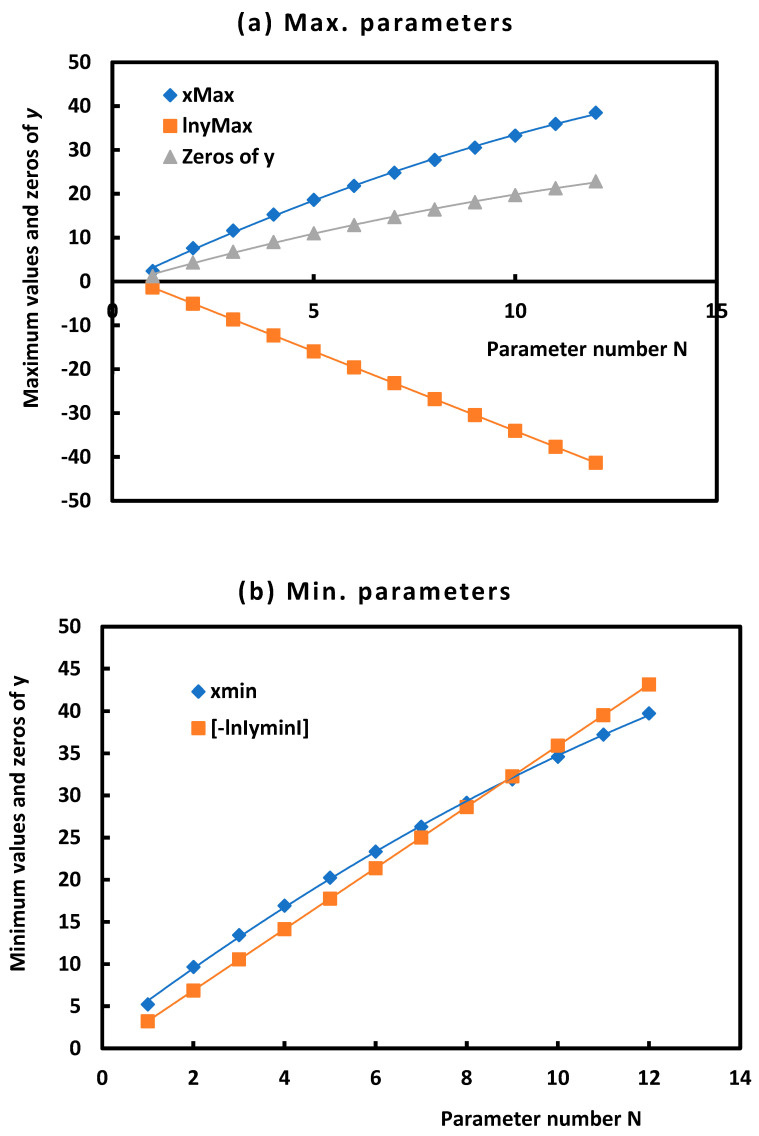
Curves of interpolation of the parameters of the grove as a function of the parameter number N. (**a**) For maximum parameters and (**b**) for minimum parameters.

**Figure 3 micromachines-14-01781-f003:**
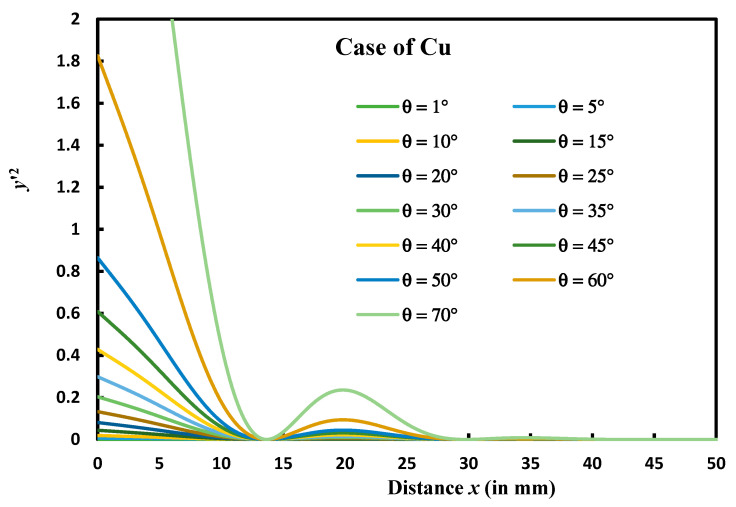
Variations in $y^{\prime 2}$ as a function of the distance *x* from the symmetrical axis of the groove at different contact angles (θ from 1° to 70 ° and *m* from 0.017 to 2.75) in the case of copper element.

**Figure 4 micromachines-14-01781-f004:**
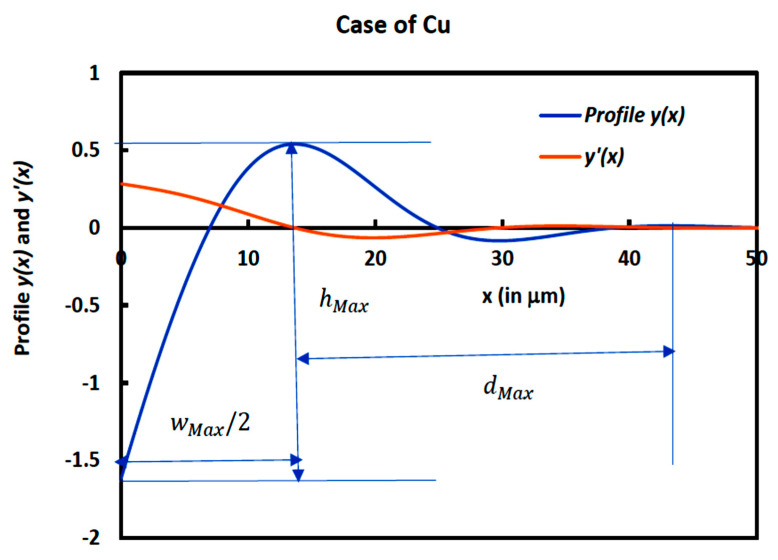
Variations in the profile *y*(*x*) *and y′*(*x*) as a function of the distance *x* from the symmetrical axis of the groove when θ = 20° (*m* = 0.364) for copper metal with the geometric characteristics.

**Figure 5 micromachines-14-01781-f005:**
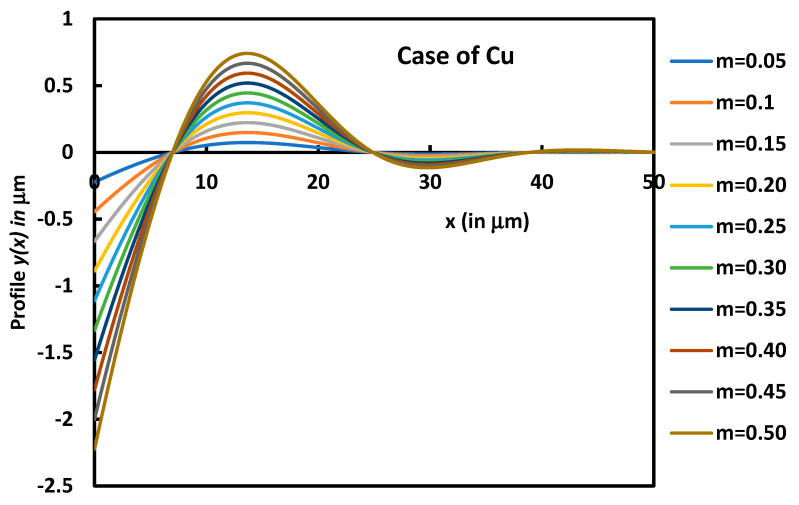
Variations in the profile *y*(*x*) as a function of the distance *x* for different values of m corresponding to θ = 2.3° to 26.6° for copper metal.

**Figure 6 micromachines-14-01781-f006:**
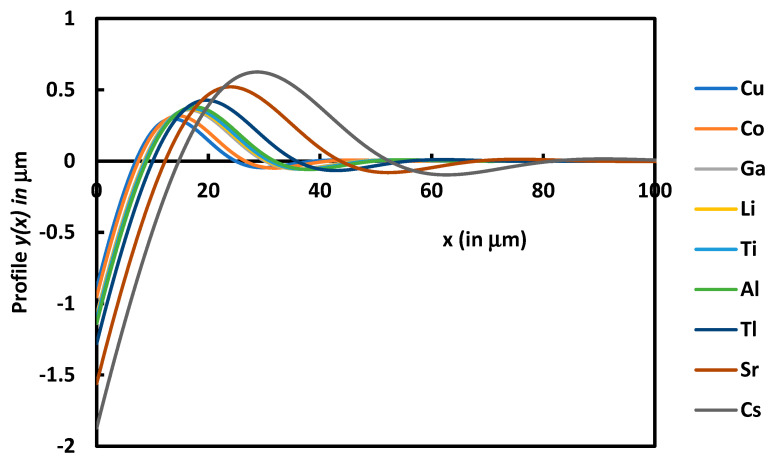
Variations in the profile *y*(*x*) as a function of the distance *x* for the different metals at t = 24 h.

**Table 1 micromachines-14-01781-t001:** Values of the coordinates of maxima and minima of the function y(x,t) with the first values of the groove shape parameters and zeros of *y*.

Number N	$x_{Max}$ $\;\mathrm{in}\;{\left(Bt\right)}^{1/4}$	$y_{Max}$ $\;\mathrm{in}\;m{\left(Bt\right)}^{1/4}$	${lny}_{Max}$	$x_{min}$ $\;\mathrm{in}\;{\left(Bt\right)}^{1/4}$	$y_{min}$ $\;\mathrm{in}\;m{\left(Bt\right)}^{1/4}$	$-ln\left|y_{min}\right|$	$x_{0}$ $\mathrm{in}\;{\left(Bt\right)}^{1/4}$ Zeros of y
1	2.4	2.60 × 10^−1^	−1.35	5.22	−4.02 × 10^−2^	3.21	1.22
2	7.62	6.44 × 10^−3^	−5.05	9.66	−1.05 × 10^−3^	6.86	4.35
3	11.62	1.70 × 10^−4^	−8.68	13.7	−2.57 × 10^−5^	10.57	6.78
4	15.26	4.50 × 10^−6^	−12.31	16.98	−7.33 × 10^−7^	14.13	9
5	18.62	1.19 × 10^−7^	−15.94	20.26	−1.95 × 10^−8^	17.76	11
6	21.82	3.17 × 10^−9^	−19.57	23.34	−5.17 × 10^−10^	21.38	12.89
7	24.82	8.42 × 10^−11^	−23.20	26.3	−1.37 × 10^−11^	25.01	14.69
8	27.74	2.24 × 10^−12^	−26.83	29.14	−3.64 × 10^−13^	28.64	16.44
9	30.54	5.94 × 10^−14^	−30.45	31.9	−9.67 × 10^−15^	32.27	18.08
10	33.26	1.58 × 10^−15^	−34.08	34.58	−2.57 × 10^−16^	35.90	19.72
11	35.94	4.19 × 10^−17^	−37.71	37.22	−6.84 × 10^−18^	39.52	21.27
12	38.5	1.11 × 10^−18^	−41.34	39.78	−1.82 × 10^−19^	43.15	22.83

**Table 2 micromachines-14-01781-t002:** Values of the differences between two consecutive maxima and minima.

Number	$∆x_{Max}$ $\mathrm{in}\;{\left(Bt\right)}^{1/4}$	$\left|{∆lny}_{Max}\right|$	$∆x_{min}$ $\mathrm{in}\;{\left(Bt\right)}^{1/4}$	$∆\left[-ln\left|y_{min}\right|\right]$
1	-	-	-	-
2	5.22	3.70	4.44	3.65
3	4.00	3.63	4.04	3.71
4	3.64	3.63	3.28	3.56
5	3.36	3.63	3.28	3.63
6	3.20	3.63	3.08	3.63
7	3.00	3.63	2.96	3.63
8	2.92	3.63	2.84	3.63
9	2.80	3.63	2.76	3.63
10	2.72	3.63	2.68	3.63
11	2.68	3.63	2.64	3.63
12	2.56	3.63	2.56	3.63

**Table 3 micromachines-14-01781-t003:** Equations of interpolation of the various parameters of the groove profile.

Parameters of the Groove	Equation of Interpolation	Linear Regression Coefficient R^2^
$x_{Max}$ in (*Bt*)^1/4^ = *f*(*N*)	$x_{Max}$ = −0.0929 N^2^ + 4.3906N − 1.1605	0.9991
${lny}_{Max}$ = *f*(*N*)	${lny}_{Max}$ = 0.0012 N^2^ − 3.6476N + 2.2688	1.0000
Zeros of y$\;\mathrm{or}\;x_{0}\;$in (*Bt*)^1/4^	$x_{0}$ = −0.0579 N^2^ + 2.6546N − 0.9316	0.9990
$x_{min}$ in (*Bt*)^1/4^ = *f*(*N*)	$x_{min}$ = −0.0767 N^2^ + 4.0748N + 1.6466	0.9996
$-ln\;\left|y_{min}\right|$ = *f*(*N*)	$-ln\;\left|y_{min}\right|$ = −0.0006 N^2^ + 3.6352N − 0.3982	1.0000
${lny}_{Max}=f(x_{Max})$	${lny}_{Max}$$=-0.0102\;x_{Max}^{2}$$-0.7048\;x_{Max}$ + 0.6885	0.9998
$x_{0}=f(x_{Max})$	$x_{0}$$=-0.0002\;x_{Max}^{2}$$+0.6073\;x_{Max}$ − 0.2429	1.0000
$-ln\;\left|y_{min}\right|=f(x_{min})$	$-ln\;\left|y_{min}\right|$$=0.0093\;x_{min}^{2}$$+0.7442\;x_{min}$ − 1.0789	1.0000
$x_{0}=f(x_{min})$	$x_{0}$$=-0.0012\;x_{min}^{2}$$+0.6723\;x_{min}$ − 2.1302	0.9999
$\mathrm{Inflexion}\;\mathrm{point}\;x_{Inf.}=f(N)$	$x_{Inf.}$ = −0.0436 N^2^ + 2.3829 N + 1.378	0.9996

**Table 4 micromachines-14-01781-t004:** Comparison between the results of our analytical solution and those obtained by Mullins.

Studied Parameter	Results Obtained by Using Our Solution	Results Obtained by Mullins
Approached equation of the groove profile	$g\left(x\right)=-0.1737\;x^{2}+0.8609x-0.7958$ $R^{2}=0.9997;\;for\;0\leq x\leq 2.40$	$g\left(x\right)=-0.288\;x^{2}+x-0.780$ for 0≤x≤1
First zero of *y*	1.22	1.14
Coordinates of the principal maximum	(2.40; 0.260)	(2.30; 0.193)
Coordinates of the first inflexion point	(3.475; 0.131)	3.43
$\mathrm{Equations}\;\mathrm{of}\;\mathrm{inflexion}\;\mathrm{point}\;x_{Inf.}=f(N)$	$x_{Inf.}$ = −0.0436 N^2^ + 2.3829N + 1.378 R^2^ = 0.9996	Not given
Positive inflexion point relation	$y_{Inf.(+)}$$=-0.0134\;x_{Inf.(+)}^{2}$$-\;0.6214\;x_{Inf.(+)}$+ 0.3252 R^2^ = 0.9999	Not given
Negative inflexion point relation	$y_{Inf.(-)}$$=0.012\;x_{Inf.(-)}^{2}$$+\;0.6638\;x_{Inf.}$ − 0.6231 R^2^ = 1	Not given

**Table 5 micromachines-14-01781-t005:** Separation distance between two consecutive maxima or minima and their ratios on the groove depth.

Separation Distance	Equation of Interpolation	Ratio d/h
Between two consecutive maxima	$d_{Max}=6.2355\times {\left(Bt\right)}^{1/4\;}N^{-0.365}$	$5.995\;N^{-0.365}/m$
Between two consecutive minima	$d_{min}=5.3909\times {\left(Bt\right)}^{1/4\;}N^{-0.305}$	$7.286\;N^{-0.365}/m$

**Table 6 micromachines-14-01781-t006:** Values of the principal maximum, distance between the two first maxima and their ratios by using our analytical solution compared to those obtained by Mullins.

Studied Parameter	Results from Our Solution	Results of Mullins
$\mathrm{Depth}\;\mathrm{of}\;\mathrm{the}\;\mathrm{groove}\;\mathrm{profile},\;h_{Max}$	$h_{Max}=1.040\times m{\left(Bt\right)}^{1/4\;}$	$h_{Max}=0.973\times m{\left(Bt\right)}^{1/4\;}$ With an error of 6.5%
Separation distance between the two first maxima	$d_{Max}=5.22\;{\left(Bt\right)}^{1/4\;}$	$d_{Max}=4.6\;{\left(Bt\right)}^{1/4\;}$ With an error of 11.88%
Ratio d/h	$\frac{d_{Max}}{h_{Max}}=\frac{5.02}{m}$	$\frac{d_{Max}}{h_{Max}}=\frac{4.73}{m}$ With an error of 5.78%

**Table 7 micromachines-14-01781-t007:** Coordinates of the positive and negative inflexion points and relations between coordinates.

**Number**	$\mathrm{Abscissa}\;\mathrm{of}\;\mathrm{the}\;\mathrm{Positive}\;\mathrm{Inflexion}\;\mathrm{Point}\;\mathrm{in}\;{\left(Bt\right)}^{1/4}$	**Ordinate of the Positive Inflexion** $\mathrm{Point}\;\mathrm{in}\;m{\left(Bt\right)}^{1/4}$
1	3.475	1.310 × 10−1
2	8.295	3.436 × 10−3
3	12.275	9.068 × 10−5
4	15.855	2.410 × 10−6
5	19.185	6.503 × 10−8
6	22.325	1.744 × 10−9
Equation	${ln\;y}_{Inf.(+)}$ = −0.0134 $x_{Inf.(+)}^{2}$ − 0.6214 $x_{Inf.(+)}$+ 0.3252; R^2^ = 0.9999	
**Number**	**Abscissa of the Negative Inflexion Point**	**Ordinate of the Negative Inflexion Point**
1	6.055	−2.109 × 10−2
2	10.355	−5.568 × 10−4
3	14.105	−1.487 × 10−5
4	17.545	−4.013 × 10−7
5	20.775	−1.040 × 10−8
6	23.845	−2.823 × 10−10
Equation	${-ln(-y}_{Inf.(-)})$ = 0.012 $x_{Inf.(-)}^{2}$ + 0.6638 $x_{Inf.(-)}$ − 0.6231; R^2^ = 1.0000	

Two expressions between coordinates of the negative and positive inflexions were given on Table 7 showing parabolic variations with excellent linear regression coefficients equal to 1.0000.

**Table 8 micromachines-14-01781-t008:** Thermodynamic parameters of *Au* and *Mg*.

Molecular mass *m*	$1.7\times 10^{-25\;}kg$
Temperature *T* (K)	725.15 K
Surface energy *γ*	1 J/m^2^
$\mathrm{Number}\;\mathrm{of}\;\mathrm{molecules}/m^{2},\;N_{S}$	$1.5\times 10^{19}molecules/m^{2}$
*kT*	10^−20^ J
$D_{S}$	10^−7^ m^2^/s
Molecular volume ω	$1.7\times 10^{-29}\;m^{3}$
Vapor pressure P_0_ of Au	$1.3\times 10^{-3}\;Pa$
P_0_ of Mg	$2.4\times 10^{2}\;Pa$

**Table 9 micromachines-14-01781-t009:** Values of *C*, *B*, and profile area of *Au* and *Mg* by using our new method compared to the values of Mullins.

Parameter	Our Results	Mullins Results
C	$2.8\times 10^{-17}P_{0}\;(in\;m^{2}/s)$	$3\times 10^{-17}P_{0}\;(in\;m^{2}/s)$
B	$4.3\times 10^{-26}m^{4}/s$	$10^{-26}m^{4}/s$
σ	$\sigma =\frac{1148.48}{P_{0}\;}t^{-1/2}$	$\sigma =\frac{2828.40}{P_{0}\;}t^{-1/2}$
$\sigma_{Au}$	$8.8\times 10^{5}t^{-1/2}$	$2.2\times 10^{6}t^{-1/2}$
$\sigma_{Mg}$	${4.8\;t}^{-1/2}$	${11.8\;t}^{-1/2}$

**Table 10 micromachines-14-01781-t010:** Values of ${\sigma t}^{1/2}$ and thermodynamic parameters of some metals, such as melting point: *T_MP_* (K), temperature of metal: *T* (K), vapor pressure at *T*: *P*_0_ (Pa), molar mass: *M* (g/mol), surface energy of metal: *γ* (J/m^2^) and atomic volume: *ω* (m^3^).

Metal	*M* (g/mol)	*γ* (J/m^2^)	*ω* (m^3^)	*T_MP_* (K)	*T* (K)	*P*_0_ (Pa)	${\sigma t}^{1/2}$
*Cu*	63.546	1.808	1.18 × 10^−29^	1358.2	2200	11,490.38	1.2 × 10^−5^
*Al*	26.9815	1.152	2.32 × 10^−29^	933.5	2000	2956.96	1.9 × 10^−5^
*Ti*	47.867	2.045	1.77 × 10^−29^	1941.2	2370	286.35	2.5 × 10^−4^
*Cs*	132.905	0.095	1.18 × 10^−28^	302.96	530	425.19	2.0 × 10^−4^
*Li*	6.941	0.524	2.18 × 10^−29^	453.7	970	294.34	1.5 × 10^−4^
*Co*	58.933	2.536	1.11× 10^−29^	1768.2	2120	303.04	3.8 × 10^−4^
*Ga*	69.723	0.991	1.96 × 10^−29^	302.96	1570	278.52	4.1 × 10^−4^
*Tl*	204.383	0.639	2.86 × 10^−29^	577.2	1070	318.79	5.2 × 10^−4^
*Sr*	87.62	0.415	5.60 × 10^−29^	1050.2	1030	1008.65	6.9 × 10^−5^

**Table 11 micromachines-14-01781-t011:** Calculated values of evaporation *C* and diffusion *B* constants from the experimental data.

Metal	$C\left(in\;m^{2}/s\right)$	$B\left(in\;m^{4}/s\right)$	${\left(Bt\right)}^{1/4}\left(in\;m\right)$ for 24 h
*Co*	5.9 × 10^−15^	1.6 × 10^−26^	6.1 × 10^−6^
*Ti*	9.6 × 10^−15^	2.9 × 10^−26^	7.1 × 10^−6^
*Ga*	1.0 × 10^−14^	2.6 × 10^−26^	6.9 × 10^−6^
*Li*	1.5 × 10^−14^	2.8 × 10^−26^	7.0 × 10^−6^
*Tl*	2.9 × 10^−14^	5.3 × 10^−26^	8.2 × 10^−6^
*Al*	1.2 × 10^−13^	3.4 × 10^−26^	7.3 × 10^−6^
*Cu*	1.7 × 10^−13^	1.2 × 10^−26^	5.7 × 10^−6^
*Sr*	2.4 × 10^−13^	1.4 × 10^−25^	1.0 × 10^−5^
*Cs*	2.8 × 10^−13^	2.7 × 10^−25^	1.2 × 10^−5^

**Table 12 micromachines-14-01781-t012:** Variations in the depth $h_{Max}$ (in m) of the groove in the case of diffusion of different metals as a function of time.

Metal	1 s	1 min	1 h	1 Half-Day	1 Day	5 Days	10 Days
*Co*	7.4 × 10^−8^	2.1 × 10^−7^	5.7 × 10^−7^	1.1 × 10^−6^	1.3 × 10^−6^	1.9 × 10^−6^	2.3 × 10^−6^
*Ti*	8.6 × 10^−8^	2.4 × 10^−7^	6.7 × 10^−7^	1.2 × 10^−6^	1.5 × 10^−6^	2.2 × 10^−6^	2.6 × 10^−6^
*Ga*	8.4 × 10^−8^	2.3 × 10^−7^	6.5 × 10^−7^	1.2 × 10^−6^	1.4 × 10^−6^	2.1 × 10^−6^	2.6 × 10^−6^
*Li*	8.5 × 10^−8^	2.4 × 10^−7^	6.6 × 10^−7^	1.2 × 10^−6^	1.5 × 10^−6^	2.2 × 10^−6^	2.6 × 10^−6^
*Tl*	1.0 × 10^−7^	2.8 × 10^−7^	7.7 × 10^−7^	1.4 × 10^−6^	1.7 × 10^−6^	2.6 × 10^−6^	3.0 × 10^−6^
*Al*	8.9 × 10^−8^	2.5 × 10^−7^	6.9 × 10^−7^	1.3 × 10^−6^	1.5 × 10^−6^	2.3 × 10^−6^	2.7 × 10^−6^
*Cu*	6.9 × 10^−8^	1.9 × 10^−7^	5.4 × 10^−7^	1.0 × 10^−6^	1.2 × 10^−6^	1.8 × 10^−6^	2.1 × 10^−6^
*Sr*	1.3 × 10^−7^	3.5 × 10^−7^	9.8 × 10^−7^	1.8 × 10^−6^	2.2 × 10^−6^	3.2 × 10^−6^	3.9 × 10^−6^
*Cs*	1.5 × 10^−7^	4.2 × 10^−7^	1.2 × 10^−6^	2.2 × 10^−6^	2.6 × 10^−6^	3.8 × 10^−6^	4.6 × 10^−6^

**Table 13 micromachines-14-01781-t013:** Variations in the width $w_{Max}$ (in m) of the groove in the case of diffusion of different metals as a function of time.

Metal	1 s	1 min	1 h	1 Half-Day	1 Day	5 Days	10 Days
*Co*	1.7 × 10^−6^	4.8 × 10^−6^	1.3 × 10^−5^	2.5 × 10^−5^	2.9 × 10^−5^	4.4 × 10^−5^	5.2 × 10^−5^
*Ti*	2.0 × 10^−6^	5.5 × 10^−6^	1.5 × 10^−5^	2.9 × 10^−5^	3.4 × 10^−5^	5.1 × 10^−5^	6.1 × 10^−5^
*Ga*	1.9 × 10^−6^	5.4 × 10^−6^	1.5 × 10^−5^	2.8 × 10^−5^	3.3 × 10^−5^	5.0 × 10^−5^	5.9 × 10^−5^
*Li*	2.0 × 10^−6^	5.5 × 10^−6^	1.5 × 10^−5^	2.8 × 10^−5^	3.4 × 10^−5^	5.0 × 10^−5^	6.0 × 10^−5^
*Tl*	2.3 × 10^−6^	6.4 × 10^−6^	1.8 × 10^−5^	3.3 × 10^−5^	4.0 × 10^−5^	5.9 × 10^−5^	7.0 × 10^−5^
*Al*	2.1 × 10^−6^	5.7 × 10^−6^	1.6 × 10^−5^	3.0 × 10^−5^	3.5 × 10^−5^	5.3 × 10^−5^	6.3 × 10^−5^
*Cu*	1.6 × 10^−6^	4.5 × 10^−6^	1.2 × 10^−5^	2.3 × 10^−5^	2.7 × 10^−5^	4.1 × 10^−5^	4.9 × 10^−5^
*Sr*	2.9 × 10^−6^	8.1 × 10^−6^	2.3 × 10^−5^	4.2 × 10^−5^	5.0 × 10^−5^	7.5 × 10^−5^	8.9 × 10^−5^
*Cs*	3.5 × 10^−6^	9.6 × 10^−6^	2.7 × 10^−5^	5.0 × 10^−5^	5.9 × 10^−5^	8.9 × 10^−5^	1.1 × 10^−4^

## Data Availability

No other data available.

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
