# Peer review of "Thermal Fatigue Effect on the Grain Groove Profile in the Case of Diffusion in Thin Polycrystalline Films of Power Electronic Devices"

_micromachines, 2023, doi:10.3390/mi14091781_

Round 1

Reviewer 1 Report

The authors present a thorough theoretical treatment of thermal fatigue effect on grain boundary groove profiles by arriving at an analytical solution to the linearized Mullins fourth order differential equation for the diffusion case. The method is applied to polycrystalline thin films in power electronics.

The paper in its current state is not fit for the format for a journal article such as Micromachines even if the article is a theoretical paper. At the current format it is very difficult to navigate.  A journal article needs to cater to the general readership and hence should keep the flow of information dissemination in an efficient manner. Long derivation of equations should not be part of the main text, and only the key equations and important assumptions and approximations should be included.  The methodology and the results are interesting and appropriate for this special issue however, the authors have to make major revisions before it can be considered for publication in this special issue of Micromachines. The following issues have to be resolved by the authors in revising their manuscript:

(1) Only key setup equations, assumptions and final result equation have to be included in the main text and the rest should be put in an appendix or supplementary section.  For example in Section 2, in setting up the Mullin’s equation for the diffusion case the authors Eq 1,2,4,5-9 should remain as they are important to set up the diffusion case. However, the authors should just say: 

“Eq 9 is the general case for the normal direction velocity, where c is the curvature defined by Eq (12), and y is the coordinate of a oint at the surface along the axis normal to the initial flat surface. Applying the following boundary conditions (show Eqn 17’) and adopting a change in variable and defining a new function g, as shown by Eq 21, we obtain the equation for the diffusion case. Please see Appendix Section 1 for the full derivation.)"

(2) Also note that Equations 2 and 3 are duplicates of each other. Remove one of them.

(3) In Section 3, The authors should follow the same idea as in comment (1).  Line 1 up to Eq 38 should remain intact, however after that the authors should simply say the following: 

“In order to arrive at the exact solution of Mullin’s problem we propose to new method in which a funtion r is introduced given by Eq 39 and Eq 40.  The treatment of these equations will lead to the discriminant delta and a value for u=uo= 2^(5/2)/3^(3/4).  Two cases arise for (1) delta>0, u>uo and (1) delta<0, u<uo. After applying the proper boundary conditions for each case and solving for the unknown problem parameters these two cases will give us two final analytical expressions for the function g (u) (The equation for g in page 19) and eventually the final closed form expression for the profile variation of the grove. See Appendix Section 2 for the detalied derivation.”

(4) The authors should format the Tables more carefully as the alignment for all of them are not visually agreeable for the reader.

(5) It would greatly help the readers if a diagram of the physical picture of the relationship of the groove proflile is with respect to the thin film section.

(6) On page 24, the discussion on the comparison between evaporation and diffussion cases should also follow the format recommended in comments (1) and (3), only keep the main key equations in the text and leave the derivation for the appendix. Put this derivation as section 3 of the appendix.

(7) The conclusion is too short and does not quantify the results and only mention the generalized results of the work. The authors should expand the conclusion to make it more comprehensive and representative of their key results.

After all these issues have been addressed could this article be considered for publication to this special issue of Micromachines.

The manuscript needs some minor spelling check and minor grammatical and style errors.

Reviewer 2 Report

In this article, a new modeling of the grain groove profile was proposed and new analytical expressions of the groove profile, the derivative, and the groove depth were obtained in the case of diffusion in thin polycrystalline films by the resolution of the fourth differential equation formulated by Mullins that supposed ?2 ≪ 1. Obtained some valuable results. However, there are many serious issues that need to be improved.

1.    At first, this work lacked practical experiments to verify the theory in the text. Thus, the comparison with others' research is meaningless and unconvincing.

2.    For research purposes, there are many kinds of electronic devices and relevant fabricating processes. These differences all impact the properties of the final materials, including the geometric profile of the grain boundary grooving and grain groove profile. The author should understand more practical material-forming processes and further define the exact scope of the study.

3.    In the formation process of two-phase metals, two metals will penetrate into each other, which is key to influencing the two-phase separation. However, I have not seen any relevant discussion in this article.

4.    Theoretical research is the guidance for practical production and further studying. It is meaningless if the theoretical research is detached from reality and lacks domain-related knowledge.

Round 2

Reviewer 1 Report

The authors have met all the requirements and addressed all the issues that this reviewer has raised. The manuscript in its revised form is in a much better shape and is acceptable for publication to this journal. This reviewer recommends that the manuscript be published in Micromachines Special Issue.

Reviewer 2 Report

The author has solved my questions.